# Nutritional Considerations in Celiac Disease and Non-Celiac Gluten/Wheat Sensitivity

**DOI:** 10.3390/nu15061475

**Published:** 2023-03-19

**Authors:** Fardowsa Abdi, Saania Zuberi, Jedid-Jah Blom, David Armstrong, Maria Ines Pinto-Sanchez

**Affiliations:** Department of Medicine, Faculty of Health Sciences, Farncombe Family Digestive Health Research Institute, McMaster University, Hamilton, ON L8S 4K1, Canada

**Keywords:** nutrition assessment, gluten, celiac disease, wheat sensitivity

## Abstract

A gluten-free diet (GFD) is the only available treatment for celiac disease (CeD), and it may also improve symptoms in non-celiac gluten/wheat sensitivity (NCGWS). In CeD, gluten triggers an immune reaction leading to enteropathy, malabsorption, and symptoms; in NCGWS, the mechanism leading to symptoms is unknown, and neither wheat nor gluten triggers enteropathy or malabsorption. A strict GFD is, therefore, necessary for CeD, but a gluten-restricted diet (GRD) may suffice to achieve symptom control for NCGWS. Regardless of this distinction, the risk of malnutrition and macro- and micronutrient deficiencies is increased by the adoption of a GFD or GRD. Thus, patients with CeD or NCGWS should undergo nutritional assessment and subsequent monitoring, based on evidence-based tools, under the care of a multidisciplinary team involving physicians and dietitians, for the long-term management of their nutrition. This review gives an overview of available nutrition assessment tools and considerations for the nutritional management of CeD and NCGWS populations.

## 1. Introduction

Celiac disease (CeD) is a chronic, multi-system autoimmune disorder triggered by gluten and characterized by small intestinal enteropathy in individuals with genetic predisposition (positive for HLA-DQ2/HLA-DQ8 alleles) [1]. The diagnosis of CeD is based on specific serological tests—such as anti-tissue transglutaminase (IgA-TG2), anti-deamidated gliadin peptide (IgG-DGP) or anti-endomysial antibodies (IgA-EmA)—and confirmed, histologically, by an enteropathy consisting of villous atrophy, crypt hyperplasia, and increased intraepithelial lymphocytes observed in duodenal biopsies [1,2]. For children, owing to the tight correlation between serology and villous atrophy, the European Society for Paediatric Gastroenterology, Hepatology, and Nutrition (ESPGHAN) endorses the option of omitting the small intestinal biopsy in children with an IgA-TG2 concentration of more than ten times the normal upper limit and a positive IgA-EmA on a second blood sample [2]. Non-celiac gluten/wheat sensitivity (NCGWS) is a clinical disorder characterized by gastrointestinal (GI) and extraintestinal symptoms induced by wheat and/or gluten-containing foods, having ruled out the diagnoses of CeD and wheat allergy (WA) [3,4]. Unlike CeD, there is no specific test to diagnose NCGWS, and the pathophysiology of this condition is not clear. In NCGWS, gluten intake does not cause enteropathy or malabsorption, but different gastrointestinal and extraintestinal symptoms, such as abdominal pain, diarrhea, constipation, bloating, headaches, and brain fog, among others, are triggered by wheat or gluten intake [4,5,6,7]. Previous studies have shown symptoms to be induced by gluten [5], while others have indicated that different components of wheat, such as fructans [6,7] or amylase trypsin inhibitors [4], can trigger symptoms in NCGWS. CeD and NCGWS are considered common conditions, with an estimated prevalence of CeD and NCGWS of 1% [2] and up to 7%, respectively [4], and the incidence is increasing over time [8].

The only available treatment for CeD is strict adherence to a gluten-free diet (GFD) [9], which is also recommended for patients with NCGWS [3]. Patients with CeD or NCGWS adopting a GFD are at risk of malnutrition, which refers to imbalance, deficiency, or excess in nutrient and/or energy intake and encompasses the subcategories of undernutrition, overnutrition, and micronutrient-related deficiencies [10,11,12,13]. Therefore, nutritional considerations play an important role in the clinical management of patients with these conditions. We have therefore conducted a review to evaluate the role of nutrition assessment, macro- and micronutrient deficiencies, and recommendations for nutritional management in patients with CeD and NCGWS adopting a gluten-free diet (GFD).

## 2. Nutritional Assessment Tools to Evaluate Nutritional Status

Nutritional assessment tools are used to determine the nutritional status of an individual or group with respect to their intake and utilization of nutrients [14,15]. This is done by evaluating dietary intake and identifying nutrient deficiencies, as well as identifying the severity of malnutrition [14]. Given the specific nutritional needs of patients with CeD and NCGWS, a nutrition assessment is recommended to provide an initial evaluation of nutritional status in clinical practice. Nutritional assessments should generally be conducted by a medical doctor, a registered dietitian (RD), or a nurse specializing in nutrition [16]. Systematic review-based guidelines from the Academy of Nutrition and Dietetics Evidence Analysis Library recommend that nutritional assessments and therapy be emphasized for patients with CeD [17]. The components of nutritional assessments are summarized in Figure 1.

### 2.1. Clinical Assessment

Several tools [18,19,20,21,22,23,24,25,26] can be used to evaluate nutritional status in patients with chronic diseases such as CeD and NCGWS (Table 1). Nutritional risk screening, a simple and rapid first-line tool to detect patients at risk of malnutrition, should be performed systematically in outpatient and inpatient settings [20]. Patients identified as being at nutritional risk should undergo a more detailed nutritional assessment to quantify specific nutritional problems. The most utilized tool for nutritional assessment is the Subjective Global Assessment (SGA), which evaluates parameters such as dietary intake, recent weight change, functional status, and body composition [18,19,20]. The SGA provides patients with a 3-grade rating of their nutritional status: SGA “A” indicates no malnutrition, SGA “B” indicates mild/moderate malnutrition and SGA “C” indicates severe malnutrition [18]. However, despite the SGA’s development to evaluate the nutrition status, due to an overreliance on BMI and weight loss for detecting malnutrition, it may overlook malnutrition in individuals who are overweight/obese [27]. If a patient with CeD/NCGWS is overweight or obese, available tools such as the Edmonton Obesity Staging System (EOSS) can be useful to assess the risk of malnutrition, which is relevant to direct nutritional recommendations in obesity. The EOSS is a staging system that goes beyond evaluating obesity using measures of BMI and waist circumference [28]. Instead, patients are classified into obesity classes based on BMI ranges, then sub-classified into stages of mortality risk using physical, metabolic, and psychological evaluations of an individual’s health [28]. Under the EOSS, obese patients are only recommended to lose weight in the higher staging categories [28,29].

### 2.2. Measurement of Body Composition

Individuals with CeD and NCGWS can have varied clinical presentations based on symptoms of weight loss and degree of malabsorption [29,30,31], which can lead to muscle wasting. Low muscle mass has been associated with falls and bone health complications, including fractures [32] and an increased risk of mortality [33]. As such, evaluating body composition is an important part of nutrition assessment. This can be done by taking anthropometric measurements during a physical exam, including measurements of arm, waist, and calf circumferences, triceps, subscapular, and sacroiliac skin folds, and weight and height [34]. The risk of developing certain health conditions can then be determined from these values. For instance, waist circumference is found to be correlated with levels of visceral fat and obesity and is a predictor of cardiovascular and metabolic risk [35]. Muscle strength is also tested in nutrition assessments using a dynamometer, which is a simple, non-invasive tool to assess handgrip strength. Handgrip strength has been found to be linked to the nutritional status of a patient [36,37], reflecting the individual’s level of physical activity [37,38]. Individuals with low handgrip strength are also likely to have poor nutritional status [36]. Other important measures of body composition include fat mass (FM), fat-free mass (FFM), and percent body fat (PBF). Untreated patients with CeD have a higher percentage of FFM than treated CeD patients, with a significant increase in PBF and weight upon adherence to GFD [39].

Common methods of evaluating body composition include dual-energy X-ray absorptiometry (DXA), air displacement plethysmography (ADP), and bioimpedance analysis (BIA) [20]. More recently, 3D body-scanning systems have been developed to provide a simple, efficient method of measuring FM, FFM, and PBF. These novel 3D body scanning tools are safe, quick, and often portable [40,41,42], offering a simple and convenient tool for use in a clinical setting. There are presently no studies that investigate body composition by utilizing 3D body scanners in the CeD and NCGWS populations. However, the use of these novel tools to evaluate FM, FFM, and PBF among patients with CeD and NCGSW may enhance the detection of health risks in nutrition assessments.

### 2.3. Dietary Intake and Measurements of Energy Needs

Assessing dietary intake is critical in the nutritional management of patients with CeD and NCGWS. Subjective assessment methods include 24-h dietary recall, dietary records, dietary history, and food frequency questionnaires (FFQs). Data are collected by an RD, a trained interviewer, or by self-report. Dietary recall and dietary history are open-ended surveys that collect a variety of information about food consumed over a specified time frame and are completed retrospectively. Data obtained include food preparation methods, ingredients used, and portion sizes [43]. While dietary recall can provide a highly detailed account of the patient’s eating habits, it is limited by its dependency on the patient’s recollections. FFQs have been used in a wide range of dietary studies and nutritional epidemiological research [43,44]. There are several validated FFQs [45,46,47] that ask patients to describe how often and how much food they ate over a specified period; usually 6 months or a year. The disadvantages of FFQs are related to recall bias, as they rely on the patient’s memory, and the time needed to complete, which can take 20–30 min on average [45]. Moreover, FFQs that do not document gluten-free (GF) foods can be inaccurate and should be reviewed by an RD before data analysis [47].

In contrast, dietary records collect dietary intake data prospectively [48]. In a study by Prentice et al. comparing the FFQ with food records and recalls, food records emerged as the best estimate for estimating energy and protein intake [44]. Due to the prospective data collection of dietary intakes in food records, they are shown to be high in both validity and precision and often serve as a reference in validation studies. However, this method is not without faults and can be limited by the patient’s modification of dietary intake to reflect social desirability. Dietary intake is difficult to measure, and all currently available methods are subject to errors and biases. As such, the strengths and limitations of each approach should be considered when selecting tools for use in clinical practice [44].

Determinations of energy needs are based on the estimation of basal energy expenditure (BEE), which accounts for 60–80% of the total energy expenditure (TEE) in healthy individuals. BEE can be assessed in two ways using: (a) predictive equations, which provide estimates of energy requirements, or (b) precise tools such as direct or indirect calorimetry to measure energy expenditure directly [49]. There are currently over 200 predictive equations [50] that can be used to estimate energy needs. The criteria for choosing relevant predictive equations are based primarily on patient characteristics, as the outcomes of different equations depend on whether the population is of normal weight, underweight, overweight, obese, critically ill, or requiring nutritional support [44,50]. The Harris-Benedict equation (HBE) and predictive equation (25 kcal/kg) have long been used to provide estimates of patient energy needs in clinical practice [50,51]. For patients with obesity, the Mifflin-St. Jeor formula [52] has demonstrated better performance. However, the accuracy of predictive equations is a concern, as differences of up to 500 kcal have been documented, depending on the population studied [53]. This amount of energy is sufficient to change weight at the rate of 0.5–1 kg a week [54], which is of clinical significance. Patients with CeD have been shown to have increased metabolic rates, potentially due to disease-induced inflammation [39], which decreases after treatment. No such studies have yet been conducted in patients with NCGWS. Therefore, the measurement of energy needs has the advantage of overestimating energy needs to provide patients with an accurate assessment of their energy requirements.

Measurement of energy needs can be obtained via direct or indirect calorimetry [49] (Figure 2). Direct calorimetry measures the thermal energy exchange produced from aerobic and anaerobic metabolism [55,56]. However, indirect calorimetry is preferable in an outpatient setting due to ease of use, relatively short duration of measurement, and lower cost compared with direct calorimetry [55]. It is therefore more relevant for use in nutritional management and long-term follow-up of patients with CeD and NCGWS. An indirect calorimetry exam determines patients’ resting energy expenditure (REE) [50]. It can also be used to determine relevant parameters such as the respiratory quotient (RQ), calculated by comparing levels of respiratory CO_2_ production with levels of O_2_ consumption [57]. The RQ value indicates substrate utilization and metabolism of carbohydrates, proteins, and fats [58], with physiological ranges between 0.67 and 1.3 [59]. RQ values can indicate whether patients are potentially underfeeding, thereby oxidizing fat stores for energy (RQ < 0.7). In contrast, the use of carbohydrate substrates for energy (RQ > 1.0) may indicate overfeeding [59]. Indirect calorimetry is a precise tool that can improve the quality of nutrition assessment and offer clinicians the opportunity to assess energy utilization and needs. Although in its early stages, it can be expected that the use of indirect calorimetry in CeD and NCGWS will help clinicians provide nutrition recommendations that can optimize patient disease management.

## 3. Nutrient Imbalance Associated with GFD

A GFD may be associated with multiple macronutrient and micronutrient imbalances, which include deficiencies and excesses.

### 3.1. Macronutrient Imbalance

The typical intake of a patient on a GFD is often low in complex carbohydrates and protein and high in fat and simple carbohydrates [11,60]. Moreover, conventional gluten-free foods have a higher caloric content, resulting in patient weight gain upon GFD adherence despite not changing the amount of food intake [11].

Complex carbohydrates in common foods such as whole-grain foods, fruits, and vegetables provide greater nutrient density than simple carbohydrates [61]. They are high in both soluble and insoluble fiber, promoting bowel motility and satiety and even improving glucose and lipid levels [62]. Several studies indicate high carbohydrate intake with low fiber [63,64,65,66] and high sugar content in gluten-free diets, due to the types of flour and starches used in GF foods.

Although dietary fat has long been controversial in nutrition and it is understood that the quality of dietary fat (saturated, monounsaturated, or polyunsaturated; processed or natural) is what influences health outcomes [66], fat quantity is also important [66]. Previous studies suggest that, in general, gluten-free food products have high saturated fat content [64,67,68,69] and hydrogenated fat content [64], although this finding is not consistent across the literature [70]. Furthermore, dietary fat has a higher energy content (9 kcal/g) than either carbohydrates or proteins (4 kcal/g) [71], and therefore, increases in dietary fat intake in a gluten-free diet can contribute to weight gain in patients adopting a GFD [11,72].

In a systematic review and meta-analysis of the dietary quality of the GFD [65], gluten-free foods were found to have low protein content. These findings are also in accordance with studies that evaluate GFD products in children and have concluded that the protein content is similarly low as found in studies in adult populations [73,74]. This is a matter of concern as protein has wide-ranging benefits that include the promotion of satiety, enhancement of thermogenesis (a component of basal energy expenditure), and maintenance of muscle mass [75].

### 3.2. Micronutrient Deficiencies Associated with GFD

Malabsorption of nutrients, particularly in active CeD, can lead to key micronutrient deficiencies, including iron, vitamin B12, and folate [76]. Furthermore, a deficiency of fat-soluble vitamins A, D, E, and K can be secondary to fat malabsorption [77]. However, micronutrient deficiencies—which include vitamins and minerals—are often present in individuals adopting a GFD, suggesting that nutrient deficiencies are not due solely to malabsorption. Micronutrient deficiencies associated with GFD can be related to the limited selection of foods and to the lack of fortification of GF products. For instance, GF products are typically low in folate [64,65] and iron [76,77]. Based on whether patients have low or borderline low blood levels of vitamins or minerals, clinicians may suggest nutritional supplementation [12]. Unprocessed foods such as fruits, vegetables, meat, and fish provide essential dietary minerals and vitamins, and therefore, CeD/NCGWS patients may benefit from GFDs based on naturally GF foods [73], as tolerated. Figure 3 summarizes the macronutrient and micronutrient imbalances associated with a GFD.

## 4. Nutritional Status of CeD and NCGWS Patients

The nutritional status of CeD populations has shifted over time, with classic clinical presentations marked by malabsorption and weight loss replaced by a pattern of weight gain and obesity [78,79]. In a recent systematic review and meta-analysis [80], only 6% of CeD patients are undernourished, while 20% are overweight or obese. Moreover, independent of the BMI category, most treated CeD patients experience significant weight increases after initiating a GFD. Patients with NCGWS had similar rates of overweight (22%), but higher rates of obesity than CeD (15% in NCGWS and 7% in CeD) [30]. Furthermore, according to a single-blind RCT, GF foods substantially reduce the thermic effect (TE) of a meal by 50% compared with whole foods and by 41% compared with processed foods, even with an isocaloric/macronutrient profile, and the subsequent reduction in metabolic rate increases the risk of weight gain and obesity [81]. Increasing overweight and obesity prevalence in the general population is also reflective of current trends in CeD/NCGWS patients, which is of concern due to the associated risk of common age-related conditions such as cardiovascular disease and dyslipidemia and the increased risk of mortality [82]. The emerging overweight/obese phenotype in CeD/NCGWS alongside the risk of obesity and weight gain on GFD highlights the need for continuous monitoring of nutritional status and weight management in these populations [82].

## 5. Nutritional Therapies for Celiac Disease and NCGWS

### 5.1. Dietary Therapies: How to Follow a Proper Gluten-Free Diet

As previously described, adhering to a GFD is the only available treatment for CeD [10], and it forms the basis for symptom management for NCGWS [77]. A strict GFD requires removing gluten, which is the most abundant protein found in wheat, rye, and barley [2,10]. Although oats are not thought to cause an immune reaction in CeD, they are often contaminated by gluten from other grains, including wheat, and therefore, only pure gluten-free oats are safe for patients with celiac disease [83]. Unlike in CeD, there is no threshold of tolerance for gluten or wheat in NCGWS.

A proper GFD would be ideally based on fresh foods that are as little processed as possible and are naturally free of gluten (see Table 2 for a list of gluten-free and gluten-processed foods). Manufactured products such as sauces, prepared soups, ice cream, sausages, candies, desserts, and fruit nectars may be contaminated by gluten, even if they are initially made from gluten-free ingredients.

To be considered gluten-free, a product should have gluten content below an established threshold. In most countries, the established threshold is less than 20 ppm or 20 mg of gluten/kg of food. This threshold is based on a study [84], which reported the effects of exposure to 0, 10, or 50 mg of gluten/day in people diagnosed with CeD. Some signs of enteropathy were found in the 50 mg/day group, whereas 10 mg/day was considered safe for most of the participants. This level is recognized internationally in the Codex Alimentarius Standard for Foods for Special Dietary Use for Persons Intolerant to Gluten (Codex Standard CXS 118–1979), which states that the gluten content of foods labeled gluten-free shall not exceed 20 ppm [85]. Gluten-free food can be identified by a gluten-free claim, a gluten-free logo, or by reading labels confirming the absence of gluten. Despite efforts by patients to avoid gluten and by manufacturers to produce GF food, adopting a strict GFD is challenging.

### 5.2. Challenges of Adopting a Strict GFD

A strict GFD is difficult to follow, and patients are often exposed inadvertently to gluten, which may explain why a significant number of CeD patients do not achieve mucosal recovery despite attempting a GFD [86]. Up to 30% of CeD patients will have persistent symptoms or persisting intestinal inflammation and are classified as having non-responsive celiac disease (NRCD). The most common causes of NRCD are gluten exposure, slow healing, concomitant conditions (e.g., bacterial overgrowth, pancreatic insufficiency, microscopic colitis, inflammatory bowel disease), or refractory celiac disease [87].

GFD adherence poses significant behavioral, social, and economic challenges for patients, especially when eating outside their homes [73]. This can negatively impact the quality of life and social activities of CeD patients. Adherence to a GFD can further pose challenges for adolescent patients who may wish to avoid exclusion in social environments, such as school. Furthermore, GFD implies dietary restrictions and a lower nutritional quality of the diet, which may lead to deficiencies of macro- and micronutrients in adult and pediatric populations [88,89]. Poor GFD adherence and/or reduced nutritional nutrient content predispose to non-recovery of nutrient deficiencies, particularly iron, leading to persistent anemia [89].

### 5.3. Nutritional Support Therapies in CeD and NCGWS

A small proportion of patients with CeD, including the 1% who have refractory celiac disease, will benefit from nutritional support therapies, such as enteral nutrition (EN) and parenteral nutrition (PN). The role of nutritional support therapies in NCGWS is less clear. Nutritional support therapies enable patients with limited ability to maintain nutritional needs by oral intake to maintain adequate nutrition for long periods of time [90]. EN is a mechanism to deliver nutrition directly to the stomach or small intestine through a feeding tube [91]. Short-term enteral access devices, such as nasogastric or nasojejunal tubes, are usually chosen for patients requiring EN for up to 4–6 weeks. For therapy lasting greater than 6 weeks in patients with an inability to tolerate oral nutrition intake in any form, more permanent options include gastrostomy or gastrojejunostomy tubes [91]. In both short- and long-term EN, close monitoring is required, as is consideration of the patient’s disease state and gastrointestinal anatomy. If the gastrointestinal tract is functional, EN should be considered before PN [91]. EN in CeD is often limited to instances where patients have severe malnutrition, for instance in refractory CeD, which is characterized by persistent enteropathy and malabsorptive symptoms despite strict GFD adherence [92].

PN bypasses the gastrointestinal tract, providing patients who have intestinal failure with intravenous (IV) nutrition temporarily or permanently through an IV catheter [93]. The use of PN has been limited to patients with refractory CeD, a subpopulation with persistent villous atrophy often leading to intestinal failure [94], including the rare patient who experiences a celiac crisis marked by metabolic disturbance and abnormal diarrhea [95]. There are currently limited studies on the use of nutritional support therapies in CeD, and, as expected, none that are specific to NCGWS. However, generally, EN is associated with lower rates of serious adverse events, such as the risk of septic complications, while being more cost-effective overall than PN [96]. Ultimately, the decision to recommend either EN or PN should be based on the individual patient’s needs.

## 6. Role of an Experienced Registered Dietitian as Part of Multidisciplinary Management in CeD/NCGWS

Poor dietary adherence can cause serious health problems, such as the increased risk of osteoporosis, bone fractures, other autoimmune conditions, and T-cell lymphoma [2]. Despite this, strict adherence to a GFD is reported by 42% to 91% of CeD patients, depending on the study method [97]. In addition, even those adhering strictly to a GFD will often consume a nutritionally inadequate GFD, rich in sugar and fat and deficient in fiber [98]. A nutritionally inadequate GFD can lead to malnutrition, overweight, and obesity, with poor outcomes in patients adopting a GFD [78,79,80]. Therefore, a registered dietitian (RD) who is experienced in the management of CeD is crucial for nutritional assessment and management, including monitoring of dietary adherence, when treating patients who adopt a GFD. There are multiple roles for an RD both at the time of diagnosis and in the follow-up, including but not limited to: (1) nutritional assessment; (2) education on the adoption of a strict GFD in CeD and appropriate gluten restriction for NCGWS; (3) ensuring nutritional adequacy of a GFD; (4) diagnosis and treatment of micronutrient deficiencies; (5) advice on adequate caloric and macronutrient intake; and (6) recommendations of nutritional support therapies, if needed. An algorithm for the nutritional assessment and management of patients with CeD and NCGWS is shown in Figure 4.

However, there are challenges with access to experienced RDs in the community, which limits people from adopting a GFD to manage nutrition [99]. Furthermore, the nutrition needs of individuals vary greatly, and thus patients require personalized nutritional recommendations that are consistent with their available resources and constraints. These recommendations should continually evolve with the patient, as assessed through regular follow-up appointments with their physician and, ideally, an experienced RD. A detailed dietary review for assessment of compliance with the diet is time-consuming (between 45 min and 1 h), expensive to the healthcare system, and limited by the lack of expert RDs. Therefore, identifying gluten immunogenic peptides (GIPs) either in stool, urine, or food is a valuable tool to assess inadvertent gluten ingestion, especially when patients are not aware of cross-contamination [98]. GIPs detect as little as 50 mg of gluten excreted for up to 5 days with very high accuracy [100]. GIPs testing can be performed as point-of-care testing and is a valuable tool to complement RD assessment of GFD adherence.

## 7. Conclusions

A gluten-free diet (GFD) remains the only treatment for CeD and the best treatment for NCGWS patients; however, a GFD has been associated with nutritional imbalance. Nutritional recommendations in CeD/NCGWS should be based on individual nutrition assessments informed by precise tools and routine measurement of macronutrient and micronutrient excesses or deficiencies common in their populations. Improving the nutritional quality of gluten-free products will likely contribute to the nutrition management of patients adopting a GFD. The classical presentation of CeD has changed, and with the increasing prevalence of overweight and obesity, alongside an increased obesogenic environment, a CeD patient can show concurrent forms of malnutrition, where they may be overnourished, while also having micronutrient deficiencies. Although there is less epidemiological data on this trend in NCGWS patients, those adopting a GFD may show a similar nutritional status to those with CeD. Future research focused on the nutritional status of patients with NCGWS will enhance our knowledge of the nutritional needs of this population. Furthermore, there is a need for a better understanding of patients’ perceptions and nutritional goals for CeD and NCGWS management, as this has not been investigated in detail. Ensuring patient access to a specialized dietitian is crucial not only for assessing GFD adherence but for the overall nutritional management of patients with CeD and NCGWS.

## Figures and Tables

**Figure 1 nutrients-15-01475-f001:**
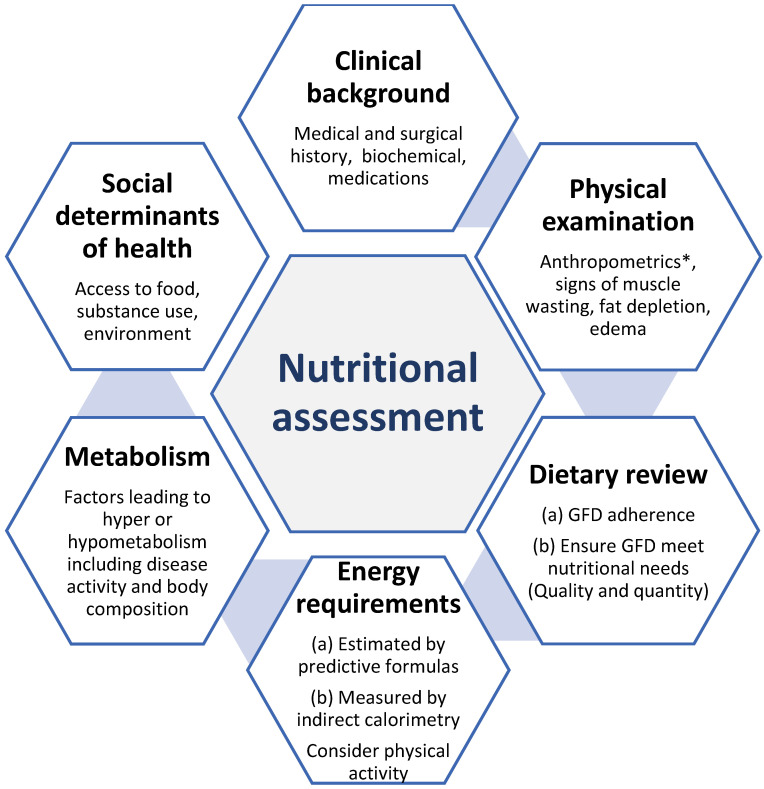
The components required in nutritional assessment to determine nutritional status in patients with celiac disease (CeD) and non-celiac gluten/wheat sensitivity (NCGWS). * Anthropometrics: height, weight, body circumferences (waist, hip, and limbs), and skinfold thickness.

**Figure 2 nutrients-15-01475-f002:**
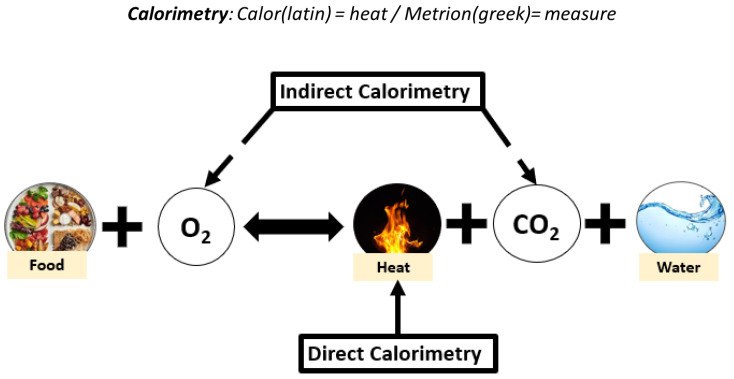
Heat, carbon dioxide (CO_2_), and oxygen (O_2_) released by food combustion can be measured by direct or indirect calorimetry to determine energy expenditure [56].

**Figure 3 nutrients-15-01475-f003:**
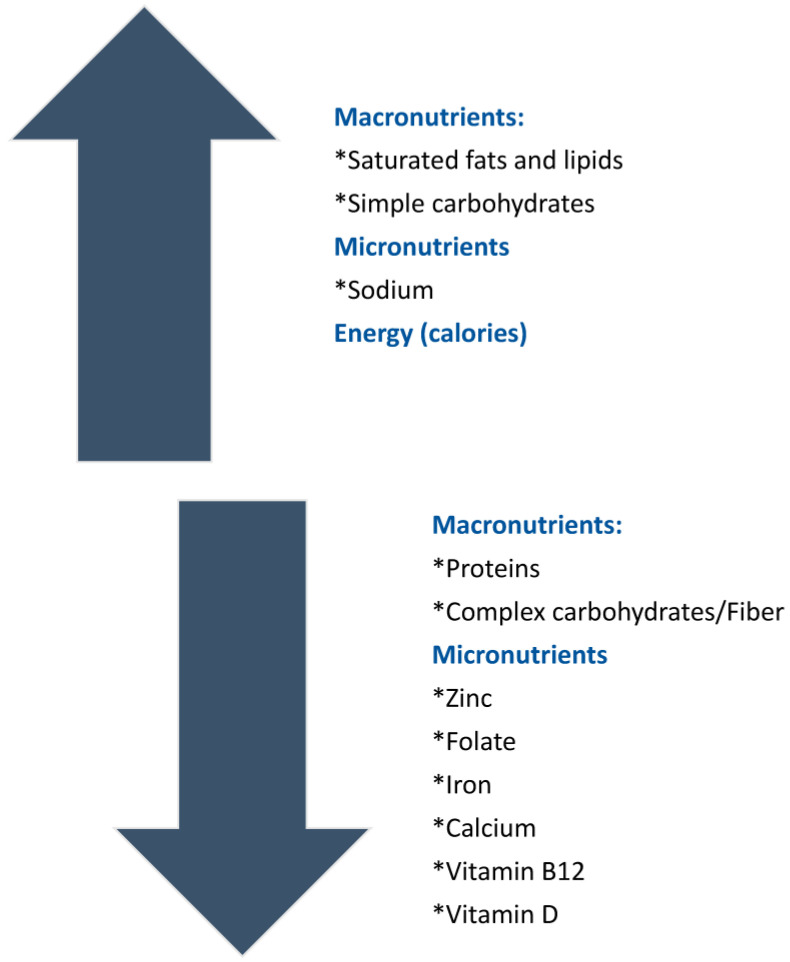
Macro- and micronutrient imbalance associated with a gluten-free diet (GFD). * Micro and micronutrients.

**Figure 4 nutrients-15-01475-f004:**
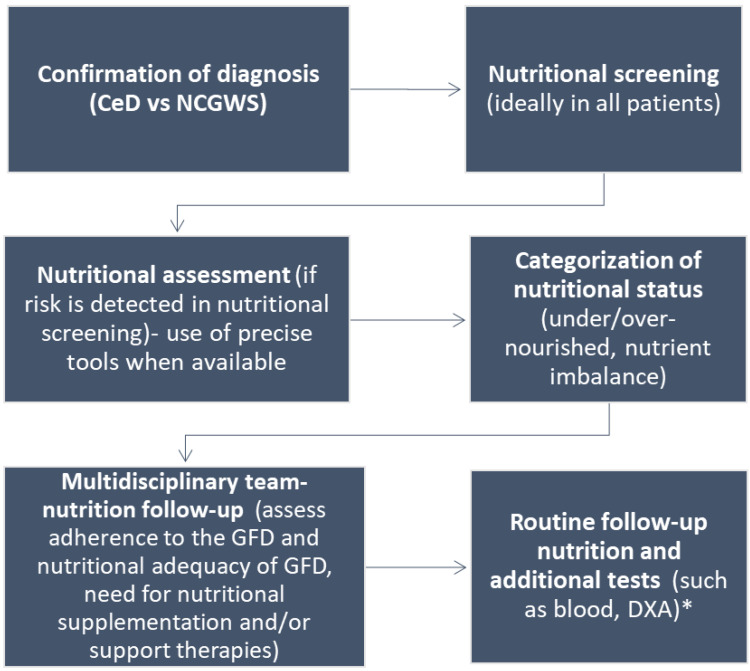
Algorithm for the nutritional assessment and management in patients with CeD and NCGWS. * Experts recommend nutrition follow-up during regular visits, usually every 6 months in the first year and then annually. The frequency of follow-up visits will be determined by nutritional status, with more frequent visits in malnourished patients.

**Table 1 nutrients-15-01475-t001:** Commonly used tools for nutrition screening and assessment available for CeD/NCGWS.

Tools	Components	Scoring	Advantages	Disadvantages
SCREENING				
Nutritional Risk Screening 2002 (NRS-2002) [20,21]	Used in an inpatient setting. Four questions in pre-screening.	If the response is positive, subsequent assessment evaluates the nutritional status and disease severity. Score ≥ 3 indicates malnutrition/malnutrition risk	Simple, well-validated tool Very reliable Can be completed within 3–5 min	Requires trained staff
Malnutrition Universal Screening Tool (MUST) [20,24]	Five-step screening tool Combines BMI score, weight loss score, and acute disease effect score to obtain malnutrition risk score.	Scores: 0 (low risk), 1 (medium risk), and ≥2 (high risk).	Malnutrition can be detected across a variety of community care settings Useful for determining the grade of malnutrition risk	Does not include low dietary intake Requires BMI and percent weight loss calculations, which can be time-consuming
ASSESSMENT				
Mini Nutritional Assessment (MNA) [20,22]	Combines nutrition screening and assessment. Covers 4 domains (nutrient intake, anthropometric measurement, global assessment, and subjective assessment).	Scores: 0–7 (malnourished), 8–11 (malnutrition risk), and 12–14 (normal nutritional status). Score ≤ 11 indicates the need for further assessment.	Quick evaluation tool No biochemical tests required Non-invasive	Useful only in limited patient populations Relies on patient self-assessment
Mini Nutritional Assessment short-form (MNA-SF) [20,23]	A short version of MNA. Covers six items (food intake, weight loss, mobility, psychological stress, neuropsychological symptoms, and BMI).	Score ≤ 11 indicates malnutrition/malnutrition risk, subsequently requiring full MNA.	Faster than complete MNA Considered as effective	Requires MNA when the patient has malnutrition risk
Subjective Global Assessment (SGA) [18,19,20]	Assessment by a healthcare (HC) provider. Seven domains (nutrient intake, weight change, symptoms, functional capacity, metabolic requirement, physical examination, and contributing factor).	Rating: SGA A (well nourished, no risk of malnutrition), SGA B (mild/moderate risk), and SGA C (severe risk).	A non-invasive and inexpensive tool Requires basic training Simple to incorporate in routine follow-ups	Only studied in some populations Does not include biochemical data Allows for subjective determination Need for physical examination
Patient-Generated Subjective Global Assessment (PG-SGA) [25]	Self-assessment by patient and assessment by HC provider. Patient-generated components (weight history, food intake, symptoms, activities, and function). HC provider component (weight loss, disease/nutritional requirements, metabolic demand, and physical exam).	The score is based on: (1) PG-SGA score = if > 4 requires intervention by a dietitian; (2) Global PG-SGA categories (stage A well-nourished; category B moderate malnutrition and stage C severe malnutrition).	Autonomy for patient Improved patient-clinician interaction Dynamic evaluation of the nutritional status	Patients may misinterpret the question Can be difficult to answer honestly The duration of recall can be long for patients
Global Leadership Initiative on Malnutrition (GLIM) criteria [26]	Framework for diagnosing malnutrition based on combinations of phenotypic (non-voluntary weight loss, low BMI, and reduced muscle mass) and etiologic (reduced food intake, disease burden/inflammatory condition) criteria.	Malnutrition is assessed based on (1) phenotypic (weight loss, low BMI, and reduced muscle mass) and (2) etiologic criteria (reduced food intake, malabsorption, and disease burden/inflammatory condition). One phenotypic and one etiologic criterion are required to define malnutrition.	High sensitivity Good performance as a screening tool	Low performance compared with SGA Low specificity False positive risk is high

**Table 2 nutrients-15-01475-t002:** Gluten-free and gluten-containing foods.

Food Group	Gluten-Free	Gluten Containing
Grains	Rice, corn, corn, tapioca, millet, sorghum, teff, buckwheat, pure gluten-free oats and quinoa	Barley, bulgar, couscous, and durum wheat and other types of wheat (einkorn, emmer, farro, kamut, and spelt (dinkel) Derived from barley (malt extract, flavoring, syrup, and vinegar) Non-pure gluten-free oats Rye, semolina, and triticale
Sugar	Sugar, honey, and sweeteners	
Meats	Fresh and frozen plain meats, offal, jerky, cured ham, and cooked ham (no flavorings), fresh and frozen fish and seafood without breading, canned, or in oil	Processed meat may contain gluten Breaded chicken, fish, or meat
Fruits and vegetables	Fresh, in-syrup, and most dried fruits (except dried figs, which may contain gluten), and vegetables	Processed fruits, jams, or vegetables flavored may contain gluten
Nuts	Raw nuts (roasted nuts may contain gluten), shelled and unshelled	Flavored nuts or mixed nuts may contain gluten
Condiments	Oil and traditional butter, vinegar	Flavored oils may contain gluten Soy sauce often contains gluten
Eggs	Eggs	Processed, scrambled, omelets may contain gluten
Hot and soft drinks	Coffee beans or ground coffee, unprocessed herbal teas, soft drinks (orange, lemon, cola, etc.), and sodas	Flavored coffees and shakes may contain gluten
Milk and dairy products	Cheeses, cottage cheese, cream, natural yogurts, and fresh curd	Processed, flavored, or mixed dairy may contain gluten
Legumes	Dried and cooked legumes in natural preserves Careful with lentils—check and remove any foreign grain if found	Processed legumes

## Data Availability

Data available from references.

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
