# Peer review of "Nutritional Considerations in Celiac Disease and Non-Celiac Gluten/Wheat Sensitivity"

_nutrients, 2023, doi:10.3390/nu15061475_

Round 1
Reviewer 1 Report (Previous Reviewer 1)
The authors responded adequately to the questions raised, with improvement of the manuscript.
Author Response
We would like to thank the reviewer for the valuable feedback provided.
Reviewer 2 Report (New Reviewer)
Dear Authors,
Verses 25-28 - notes on this part. The diagnosis of celiac disease is different in children and adults, and this has not been adequately separated.
In verses 29-34, the authors used a not entirely clear division regarding the clinical manifestation of CD, NCGS or gluten allergy. In this section, it is worth noting that very often clinically all these disorders have a similar picture (see Sharma review 2020).
In my opinion, there is no clear division between gluten-dependent disorders (CD, gluten allergy, non-celiac gluten sensitivity) and wheat-dependent disorders (and list other factors than gluten affecting the patient's condition).
Verse 140 - wrong number of Figure.
Throughout the work, there is no reference to current guidelines for celiac patients with GFD, e.g.: Mearin et al. 2022 ESPGHAN position paper on management and follow-up of children and adolescents with celiac disease. It seems reasonable to include recommendations, e.g. in the part concerning nutritional deficits, especially iron deficiency.
Author Response
Please see below our responses to the additional comments from the reviewer:
Verses 25-28 - notes on this part. The diagnosis of celiac disease is different in children and adults, and this has not been adequately separated.
R= Thank you for this comment. We have added the following sentence to reflect the non-biopsy approach in pediatric population using ESPGHAN criteria. "For children, owing to the tight correlation between serology and villous atrophy, the European Society for Paediatric Gastroenterology, Hepatology and Nutrition (ESPGHAN) include the option of omitting the small intestinal biopsy in children with an IgA-TG2 concentration of more than ten times the normal upper limit, and a positive IgA-EMA on a second blood sample (2)" (Reference Husby et al, JPEN 2020)
In verses 29-34, the authors used a not entirely clear division regarding the clinical manifestation of CD, NCGS or gluten allergy. In this section, it is worth noting that very often clinically all these disorders have a similar picture (see Sharma review 2020).
R= Thank you for this comment. We have rephrased as follow: "In NCGWS, gluten intake does not cause enteropathy or malabsorption, but different gastrointestinal and extraintestinal symptoms such as abdominal pain, diarrhea, constipation, bloating, headaches, brain fog within others, are triggered by wheat or gluten intake"
In my opinion, there is no clear division between gluten-dependent disorders (CD, gluten allergy, non-celiac gluten sensitivity) and wheat-dependent disorders (and list other factors than gluten affecting the patient's condition).
R= We agree with the reviewer there is no clear division between gluten or wheat triggers in NCGS or NCWS, and for this reason, we are using the term NCGWS which includes both of them. We have summarized the potential triggers in lines 39-43.
Verse 140 - wrong number of Figure.
R= Thank you very much for this comment. We have adjusted the figure which now reads Figure 3.
Throughout the work, there is no reference to current guidelines for celiac patients with GFD, e.g.: Mearin et al. 2022 ESPGHAN position paper on management and follow-up of children and adolescents with celiac disease. It seems reasonable to include recommendations, e.g. in the part concerning nutritional deficits, especially iron deficiency.
R= We have rephrased the sentence in lines 202-207 and added reference 90 (Mearin et al 2022 ESPGHAN) as suggested by the reviewer, and now the sentences read as follow: "Furthermore, GFD implies dietary restrictions and lower nutritional quality of the diet, which may lead to deficiencies of macro and micronutrients in adult and pediatric populations(89,90). Poor GFD adherence and/or reduced nutritional nutrient content predispose to non-recovery of nutrient deficiencies, particularly iron leading to persistent anemia (90)".
All changes are reflected in the highlighted area of the attached manuscript. Thank you very much!

Round 2
Reviewer 2 Report (New Reviewer)
Dear Authors,
Thank you for making all the changes. I accept the article in present form. I wish you success in your future scientific work.
This manuscript is a resubmission of an earlier submission. The following is a list of the peer review reports and author responses from that submission.
Round 1
Reviewer 1 Report
In this article, Abdi and colleagues aimed to review the nutrition assessment tools and the nutritional management of celiac disease patients and non-celiac gluten/wheat sensitivity populations.
However, some issues need to be addressed:
1 In the sentence “…and confirmed by the presence of villous atrophy in duodenal biopsies.” please add “crypt hyperplasia and intraepithelial lymphocytes”.
2 In table 1 there should be a column with the parameters evaluated in each tool and another column with the scale and cutoffs.
3 It is important to emphasize, particularly in the abstract and introduction, that non-celiac gluten/wheat sensitivity is not associated with severe malabsorption and that a less strict gluten-free diet may be sufficient compared to those with celiac disease.
Reviewer 2 Report
In the first part of this review the Authors described the available tools to assess nutritional status, body composition, dietary intake and energy needs. In the second part, they describe nutrient imbalances of patients with gluten-related disorders and its treatment.
There is an evident lack of link between the two parts: there is a lack of suggestions on the best test to adopt in gluten-related disorders but also on the application in previous papers. Moreover, EOSS, the only tool commented on, is not included in the Table 1. In a global vision of the paper, the first part should be very shortened and the second part more complete and exhaustive.
Para 3.1 and 3.2 should be extended: very few informations are included.
The section dealing with nutritional treatment should be more complete and should contain all the suggestions to teach patients how to correctly adher tyo a gluten-free diet.
There is a huge need of dedicated dieticians and both information and tricks should be provided. The content of the para 6 could be part of a review on any disease.
Enteral and parenteral nutrition are needed in complicated CD patients: consequently, it is conceivable the Authors did not find papers on their use in NCGWS patients.